# BIM-Based Method for the Verification of Building Code Compliance

**Fernanda Schmitd Villaschi** [1,*], **José Pedro Carvalho** [2,3] and **Luís Bragança** [2,3]

1  FSV Projetos, Vila Velha 291010-010, Brazil
2  Institute for Sustainability and Innovation in Structural Engineering (ISISE), University of Minho, 4800-058 Guimarães, Portugal; jpcarvalho@civil.uminho.pt (J.P.C.); braganca@civil.uminho.pt (L.B.)
3  Civil Engineering Department, School of Engineering, University of Minho, 4800-058 Guimarães, Portugal
*  Correspondence: fsvprojetos@gmail.com; Tel.: +55-27-99299-2400

**Abstract:** Urban planning is a valuable tool for growth control and city development, both to maintain the local urban identity and provide life quality for inhabitants. To regulate it, local governments have defined standards for a proper city's growth through municipal and detailed urban plans. Such instruments identify a set of rules which constructions and buildings must fulfil for a careful and smooth integration within the urban areas. New projects are required to comply with such rules, and designers must adapt and guarantee that their projects fulfil all the local urban requirements. However, the verification process of construction projects is still a manual procedure and often a time-consuming process, with high possibilities for inaccurate measures. Thus, this paper aims to streamline the verification procedure of construction projects' code compliance to enhance project design efficiency and save designers time. To do so, the Building Information Modelling (BIM) method will be used through the Dynamo programming software. By creating a Dynamo routine to check the building code and urban plan compliance from Brazilian municipalities, specific BIM models will be automatically analysed to detect and evaluate if it is according to local urban legislation. Results have provided a real-time decision support tool, where designers can assess if their buildings are complying with local urban codes at any time of the design stage, making it easy to innovate and integrate innovative design options, as well as to precisely communicate their buildings' code compliance. Such a method can also support municipality authorities to verify project compliance, reducing assessment and calculation errors, as well as the required time and bureaucracy for project appraisal.

**Keywords:** BIM; rule automation; parametric design; design process; urban design; urban indexes



## 1. Introduction

Building design and construction processes are usually oriented by several regulations and guidelines. The requirements for those regulations are constantly evolving and include a set of data that must be analysed to verify construction compliance. Automated design review or automated code checking is a process or system that evaluates the design based on its objects, attributes and relationships without modifying the design itself [1]. The topic has been addressed since the 1960s, following the introduction of mandatory regulations in the building industry and is becoming increasingly important with the emergence of BIM [2,3]. Without system automation, compliance checking is usually manually conducted by designers and local authorities, which is still the case in many locations. As the complexity of the designs increases [2], as well as the number of complex building codes for distinct types of constructions [4], manual compliance analyses are very time-consuming and require a deep knowledge of the building, often leading to many assessing and/or calculation errors [5,6].

BIM can be defined as "*a digital representation of a building, an object-oriented 3D model or repository of project information to facilitate interoperability and exchange of information with*

*related software applications*" [6]. Moreover, BIM is a working methodology that allows managing all the project design and data in a virtual environment during the project life cycle [7]. It creates the opportunity to virtually construct and simulate the building performance before the construction itself [8]. Some of the main benefits include constant communication among stakeholders, early detection of errors and incompatibilities, supporting decision-making and optimising costs and time [9–12]. Inherent to this methodology is the development and characterisation of a virtual model—the BIM model—which is created with object-oriented parametric modelling and is characterised by the level of development (LOD). The LOD ranges from 100 to 500 and it describes the model content and reliability [2].

With the emergence of BIM, novel approaches have been developed for automated code checking, creating better and more comprehensive procedures. Usually, automated code checking follows 4 different tasks [13]:

1. Rule interpretation—Interpretation of the requirements and translation to computer-processable rules.
2. Building model preparation—Creation and characterisation of a digital BIM model.
3. Rule execution—Execution of the established rules, usually using text format coding (Python or C#) or visual programming language (VPL) through Dynamo or Grasshopper.
4. Rule check report—Final result with building evaluation.

Following the successful results from this approach [14,15], this procedure was also adopted to conduct this research. For rule encoding, VPL was used through Dynamo software, as it is more transparent and easier to understand, especially for architecture engineering and construction stakeholders, which usually have limited knowledge of information technologies.

*Legislation*

Usually, buildings must comply with several rules defined in the local urban master plans, local construction codes and/or accessibility standards, depending on the building type, use and location. Such rules, exemplified in Figure 1 for the Brazilian case [16], are later analysed and verified by city halls, which certifies whether the building can be built in that location, with its specified characteristics and with the purpose it was designed for. If not, local authorities are demanded to send back the project, requiring the fulfilment of all local regulations.

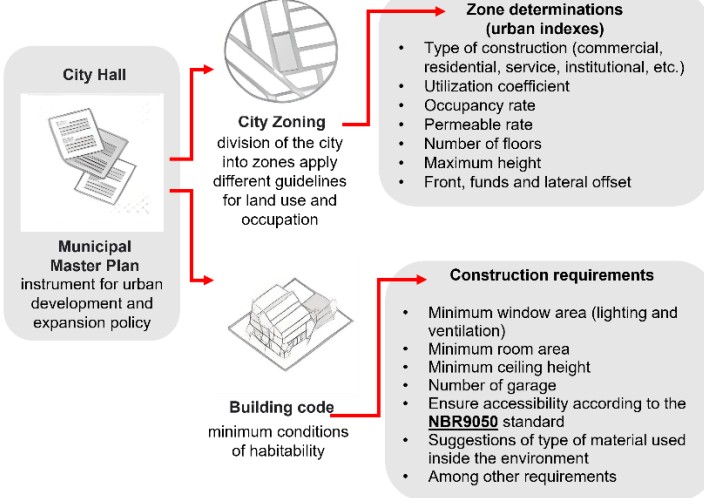

**Figure 1.** Brazilian legislation.

In Brazil, the urban master plan is the basic instrument for the urban development and expansion policy. It comprises a set of guiding principles and rules for architects and engineers to plan quality and comfortable construction for citizens.

Adopting the Brazilian specific case, the master plan establishes several zones to impose different urban indexes for an orderly use of that territory, according to the construction site location and building use. These zones are studied, evaluated and changed according to the city's expansion and use. After identifying which land zone is the building in, it is required to analyse which kinds of buildings and characteristics are allowed for the referred region. Master plans define both the building types and uses (such as commercial, institutional, residential and mixed, among others) for each identified zone. Depending on the building use, the master plans also identify a set of specific urban indexes that must be followed to guarantee safe and comfortable constructions. These indexes are usually defined by national and local governments and are mandatory for any project approval. Urban indexes are composed of different rules to aid and regulate constructions, both to provide a better quality of life for its inhabitants, as well as to ensure a correct framing of buildings with their surroundings. According to the basis for the Brazilian master plans, the following indexes must be addressed for all the municipalities:

- Utilisation coefficient: represents the relationship between the built-up area and the site area. It indicates the maximum amount of gross square metres that can be built.
- Occupancy rate: is the percentage of the site that can be occupied by any kind of construction, including the building's projection.
- Permeable rate: corresponds to the construction site permeable area percentage, to ensure proper and natural soil permeability. This permeable area must ensure the water reaches the groundwater table without any barrier during its path.
- Number of levels: maximum number of useful levels that the building can have in a specific location.
- Maximum height: represent the maximum limit for the building height. It is calculated through the relation of the distance between the street level and the highest point of the building.
- Building offset distances are also mandatory and are often seen as setbacks for the building implementation:
- Front offset: represents the maximum distance between the building's front façade and its parallel street.
- Back offset: represents the maximum distance between the building's back façade and its parallel street.
- Lateral offset: represents the maximum distance between the building's lateral façades and its parallel streets.

Together with the city zoning, urban master plans also contain the building code, which defines rules to organise the internal spaces of cities and the requirements for building habitability. These rules are valid for both new and existing buildings and aim to provide a healthy room environment for users. Some of the most commonly established requirements are:

- Minimum window area: minimum window area to ensure proper lighting and ventilation.
- Minimum room area: minimum room dimensions to serve its purposes and functionality.
- Minimum ceiling height: minimum distance between the floor and ceiling of a room.

All of the presented indexes are applied in every Brazilian city with small variations on its limits, depending on the city characteristics and building type but are always mandatory for project approval.

Another set of requirements concerns the accessibility standards for Brazil, namely the NBR9050 [17], which provides criteria and parameters for installing equipment and adapting spaces to enhance users' accessibility. To establish these criteria and technical parameters, different mobility conditions and environment perceptions are considered, including the use of assistive devices (such as prostheses), support equipment, wheelchairs,

tracking canes, assisted listening systems or anything else that may complement human individual needs. This standard aims to provide an autonomous, independent and safe use of the environment, buildings, furniture, equipment and urban elements to the greatest number of people, regardless of age, height or mobility limitation or perception. Technical service areas, or restricted areas such as engine rooms, technical passages, barrels, etc., are not required to fulfil accessibility requirements. On the other hand, multi-family residential buildings, condominiums and townhouses are required to have accessible common spaces, according to the NBR9050. Accessible autonomous units must be located on accessible routes. As an example, one of the criteria which defines room accessibility is the ramp slope, which has different requirements for pedestrians and vehicles.

To verify the compliance of all these requirements, there is a set of procedures to approve the building project. Given the number of criteria and the increased complexity of building design, these requirements usually take a significant amount of time to be assessed, as well as require multi-disciplinary knowledge about the building. Designers must guarantee that their projects fulfil all the requirements prior to submission, and city halls must verify project compliance to issue building permits. These similar tasks are usually associated with a manual calculation procedure, requiring extra time both during the design and approval stages, to verify the same indexes. The problem gets more notorious when concerning complex and large projects, where the indexes assessment takes a significant amount of time, causing several delays during the project life cycle. To overcome such an issue, the opportunity arises for process automation, both for designers and city hall experts. The automatic calculation of urban indexes can significantly reduce the required assessment process time, as well as avoid duplicate work, bureaucracy and human errors in assessing data and performing calculations. Moreover, a real-time assessment could also support decision-making during the design stage for faster achievement of an optimal building design. This need for automation allied with the recent city hall demand for innovative systems to reduce the bureaucracy of construction processes [18], such as the integration of BIM in the submission process, highlights the potential contributions of a BIM-based automated code compliance analysis. Therefore, the research question of this study regards the development of an innovative BIM-based method to automate code compliance assessment of Brazilian buildings' virtual models. Such a method will minimise time and avoid errors when assessing a building project's compliance with local and national regulations, as well as support the project design process. The Brazilian context will be considered, and a routine will be created in Dynamo to gather and process building data from BIM models for real-time automatic verification of building code and urban indexes compliance for different Brazilian cities.

## 2. Materials and Methods

To reach the intended research goals, the specific case of Brazil's master plans and two different municipal regulations will be considered—Vila Velha and Florianópolis. Currently, the Brazilian submission process of building projects is analysed, verified and approved by each city hall. Projects must be submitted in a digital format, avoiding the need to be handled by different instances, but the project verification procedure is still a manual process. Each municipality analyst or expert must manually verify each project item according to the current legislation, spending a significant amount of time, delaying the licensing procedure and creating room for calculation and misunderstanding errors. Together with the need to provide a real-time decision tool for designers, this research aims to develop an automated assessment methodology to verify the master plan and building code compliance of building projects, through the use of BIM models.

### 2.1. Methodology

To accomplish the research objectives, an automated routine will be developed and applied to two different building case studies. The methodology is divided into five different stages, as presented in Figure 2.

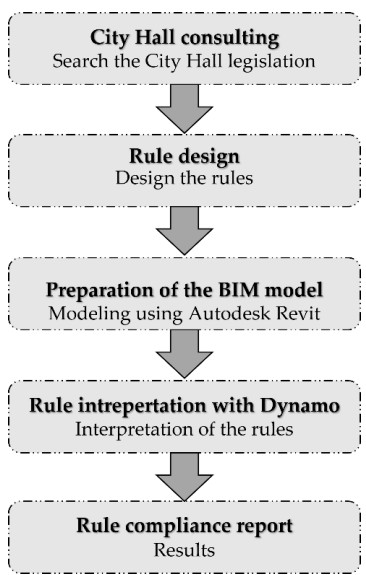

**Figure 2.** Research methodology.

The first stage will consist in collecting and identifying all the applicable legislation for buildings, both in Vila Velha and Florianópolis, Brazil. This will be made by consulting the city hall's regulations to further understand the municipal master plan requirements for the case studies building types and locations. Then, all the collected data will be carefully organised and analysed to clearly identify and assess mandatory rules.

During the following stage—rule design—the identified requirements from the previous stage will be theoretically designed to match the software code, as they are originally produced in human language format and must be interpreted and translated for software codification, in order to create a more conducive format for rule checking [19].

The next step will be the building model's creation. To do so, the BIM platform Autodesk Revit will be used to model and characterise the case studies. This authoring platform was chosen as it is the most used among researchers in the field [20,21]. It also offers the possibility to develop personal interfaces through Dynamo, which is the purpose of this study. The adopted case studies are presented in the following section. To conduct the modelling, an Autodesk Revit template was used, containing predefined schedules of room-type keys, windows and rooms, as well as settings and parameters to facilitate usability. Such definitions allow for a faster data collection from Dynamo, in order to quickly proceed with the analysis.

With the BIM model created, the Dynamo routine will be properly developed both to gather the required data from the models, as well as to perform the analysis calculations. Then, the results for each index will be faced with the master plan and local requirements to check if the building is complying with local standards and if a construction license can be issued.

Finally, the Dynamo routine will be performed to assess understudy indexes and a compliance report will be produced, indicating which indexes are being properly complied with and if the project can be approved or not.

### 2.2. Case Studies

To prove the concept and apply the developed method, two different case studies have been selected in two different Brazilian locations. The aim is to prove the method's applicability and functionality for different building types with distinct local requirements.

Thus, the first case study (Figure 3) consists of a one-level single-family residential building (SF building) located in Vila Velha, Brazil. It has some of the most representative characteristics of Brazilian houses—a detached single-family house, with a colonial ceramic tile roof and ceramic brick masonry walls. The house has 2 bedrooms, a kitchen, laundry,

dining room, living room, balcony and garage (each compartment area is described in Figure 3). The construction is located on a plan site of 280.00 m² and has a gross construction area of 72.23 m², a permeable area of 141.27 m² and a projection area of 100.99 m². The distances between the building and its limits are also presented in Figure 3, where it is possible to identify the 8.00 m front offset, the 3.35 m back offset and the lateral offset, which ranges between 2.80 m and 2.85 m. Concerning the ceiling interior height, it ranges between 240.00 cm and 270.00 cm for interior areas, while the garage has a height of 300.00 cm. The total building height is 4.05 m, while the total window area is 9.12 m², corresponding to 12.63% of the floor area. To access the building, there are two different ramps—one for vehicles (garage) and another for pedestrians—with 1.39% and 8.33% respectively.

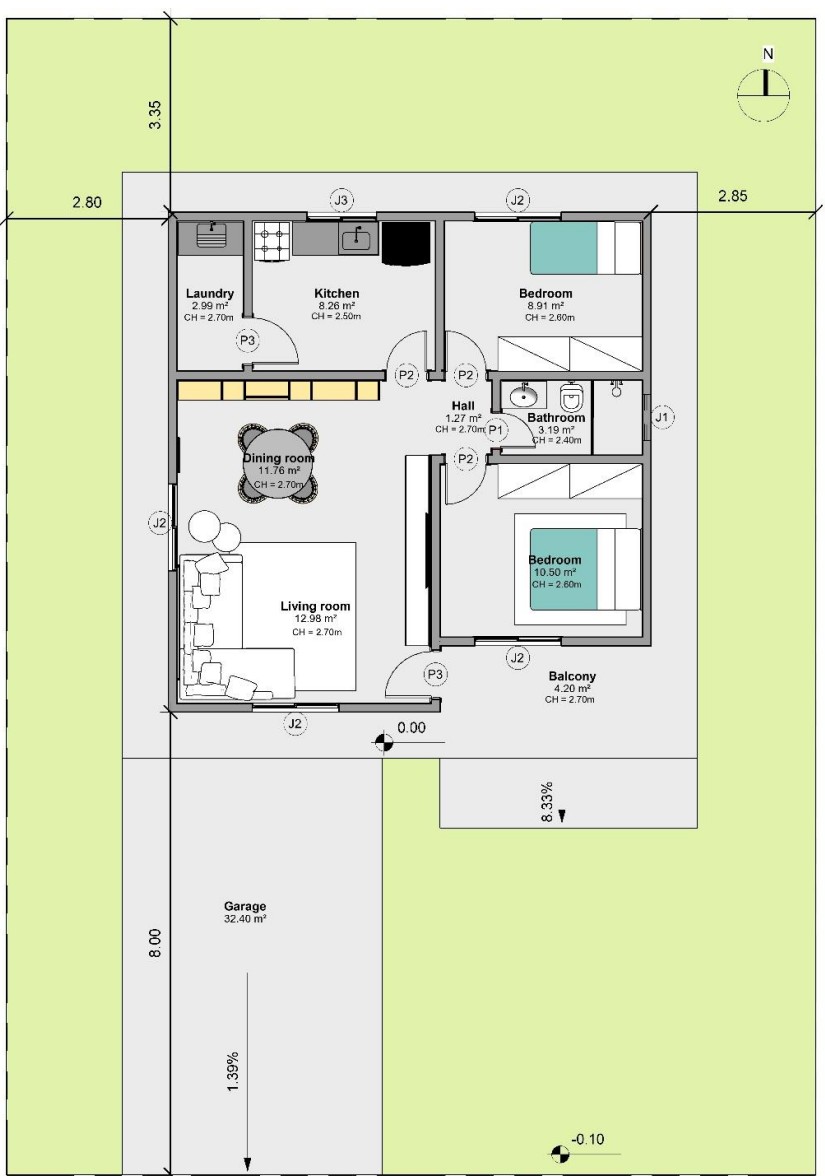

**Figure 3.** Single-family case study floor plan (in metres).

For this project to be approved by the Vila Velha city hall, it must comply with the norms and rules defined in Vila Velha's Municipal Master Plan, for this building typology—single-family residential buildings—and location—Priority Occupation Zone 03 (one of the city preferential zones for residential occupation).

The second case study (Figure 4), is a multi-family residential building (MF building) with 7 levels, located in Florianópolis, Brazil. The building is inserted in an 800.00 m²

site and has a total gross construction area of 1732.56 m$^2$, a permeable area of 334.99 m$^2$ and a projection area of 303.50 m$^2$. The building is intended for residential use and has a service area on the 7th floor, as well as an entrance and common areas on the ground floor. The building has also a basement for garage purposes, which is accessible through a vehicle ramp with a 15.00% slope. The building has a total of 10 dwellings and a total height of 20.30 m. Regarding the building implementation, it has a 4.12 m front offset, a 7.18 m back offset and a lateral offset ranging between 3.80 m and 5.15 m. Each residential level is composed of two dwellings, as presented in Figure 4, with approximately 122.76 m$^2$ of floor area each. Each apartment is divided into several rooms: two bathrooms, three bedrooms, a kitchen, hall, laundry, living room and a balcony. The room's interior height is 270.00 cm for every dwelling, which has a window area of 14.76 m$^2$, corresponding to 12.02% of the floor area.

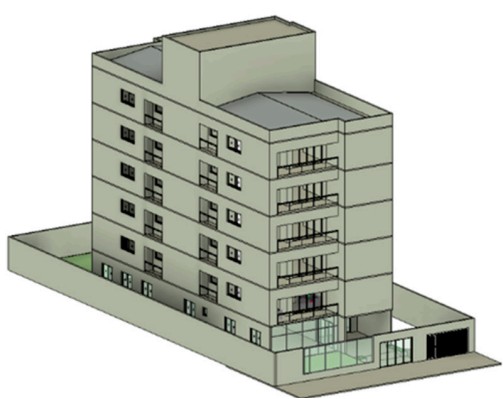

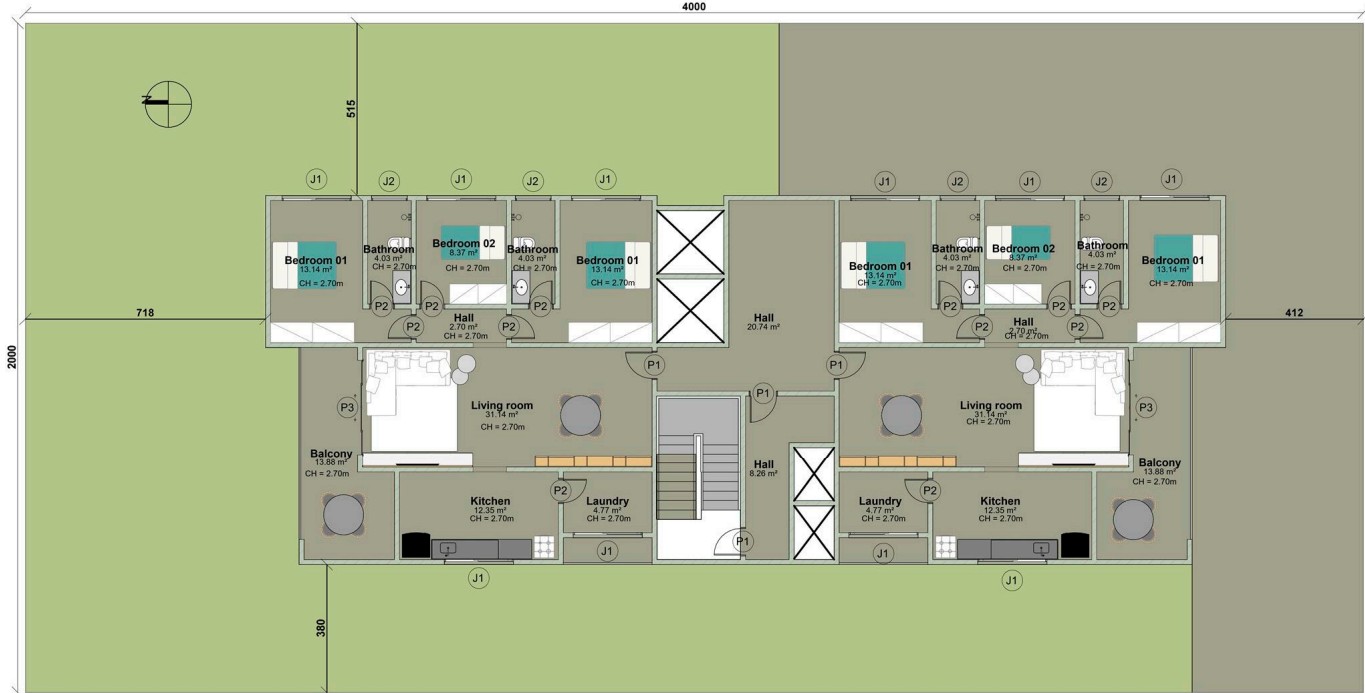

**Figure 4.** Multi-family case study.

For approval, this project must comply with all Florianóplis building code and master plan standards for this building typology—multi-family residential building—and location—AMC 12.5 Zone (central mixed area).

## 3. Results

Following the adopted methodology, the first step was the legislation data collection from the municipalities. By consulting the urban master plan, the requirements presented in Table 1 were identified for residential buildings located in the Priority Occupation Zone 03 (Vila Velha) and located in the AMC 12.5 zone (Florianópolis).

**Table 1.** Vila Velha and Florianópolis master plan requirements.

| Requirements | Vila Velha | Florianópolis |
|---|---|---|
| Maximum utilisation coefficient | 3.50 | 4.80 |
| Minimum front offset | 3.00 m | 4.00 m |
| Minimum back offset | 3.00 m | 1.50 m |
| Minimum lateral offset | 1.50 m | 1.50 m |
| Maximum number of levels | 15 floors | 10 floors |
| Maximum building height | 47.00 m | 45.00 m |
| Maximum occupancy rate | 60.00% | 50.00% |
| Permeable rate | Min. 15.00% | Max. 70.00% |
| Maximum slope for a pedestrian ramp | 8.33% | 8.33% |
| Maximum slope for a vehicle ramp | 20.00% | 20.00% |

When analysing the remaining local standards, namely the building code of each municipality, there are other indexes and rules that the building must comply with: minimum ceiling height; minimum compartment area; minimum window area for ventilation and lighting. These minimum limits are the same for both municipalities and are defined in each building code [22,23], respectively, for the different compartment types, according to Table 2. The local regulation defines both the minimum area and ceiling height for different compartments, while the minimum window area depends on the compartment floor area. For this specific case, the building code limits are the same for both locations.

**Table 2.** Building code minimum requirements in Vila Velha and Florianópolis.

| Minimum Requirements | Hall | Service Bathroom | Social Bathroom | Living Room | Kitchen | Pantry | Laundry | Garage |
|---|---|---|---|---|---|---|---|---|
| Area (m$^2$) | 1.00 | 1.60 | 2.50 | 10.00 | 4.50 | 1.60 | 1.60 | 10.35 |
| Window area | - | 1/8 | 1/8 | 1/6 | 1/8 | 1/10 | 1/10 | 1/20 |
| Ceiling height (cm) | 230.00 | 230.00 | 230.00 | 260.00 | 230.00 | 260.00 | 230.00 | 230.00 |

During the modelling stage, there are a set of guidelines that must be followed to ensure Dynamo routine functionality:

- A personalised template should be used, which already contains the required building code parameters that must be analysed.
- Rooms must be created and characterised for every building compartment.
- Walls must be segmented in every intersection, as the routine will specifically assess the building rooms and identify the associated elements.
- Roofs must be modelled under the "Roof" category, as the building height will be evaluated through the highest point of the roof.
- Users cannot model elements using the "model in place tool", as the Dynamo routine will not consider the created categories.
- The user must identify the site area, the construction area and the permeable area using the "area plan" function under the Autodesk Revit Architecture tab.

The personalised template was created to automate the aggregation and collection of the required building data for the local code analysis. By organising the information through means of tables, the template provides all the input data for the Dynamo routine. For each room/compartment, it is identified whether the room should have a window, the minimum area for each type of room, the minimum window area for an adequate natural

lighting and ventilation and the minimum ceiling height (considering the height from the floor finishing to the room's ceiling). This process was automated within the Autodesk Revit template, following the procedure presented in Figure 5. The model rooms are turned into a list, which has sub-lists of information for each room, including the required data for the analysis. These data are then used to verify if a specific room is complying with the understudy rules by facing its characteristics with the building code standards.

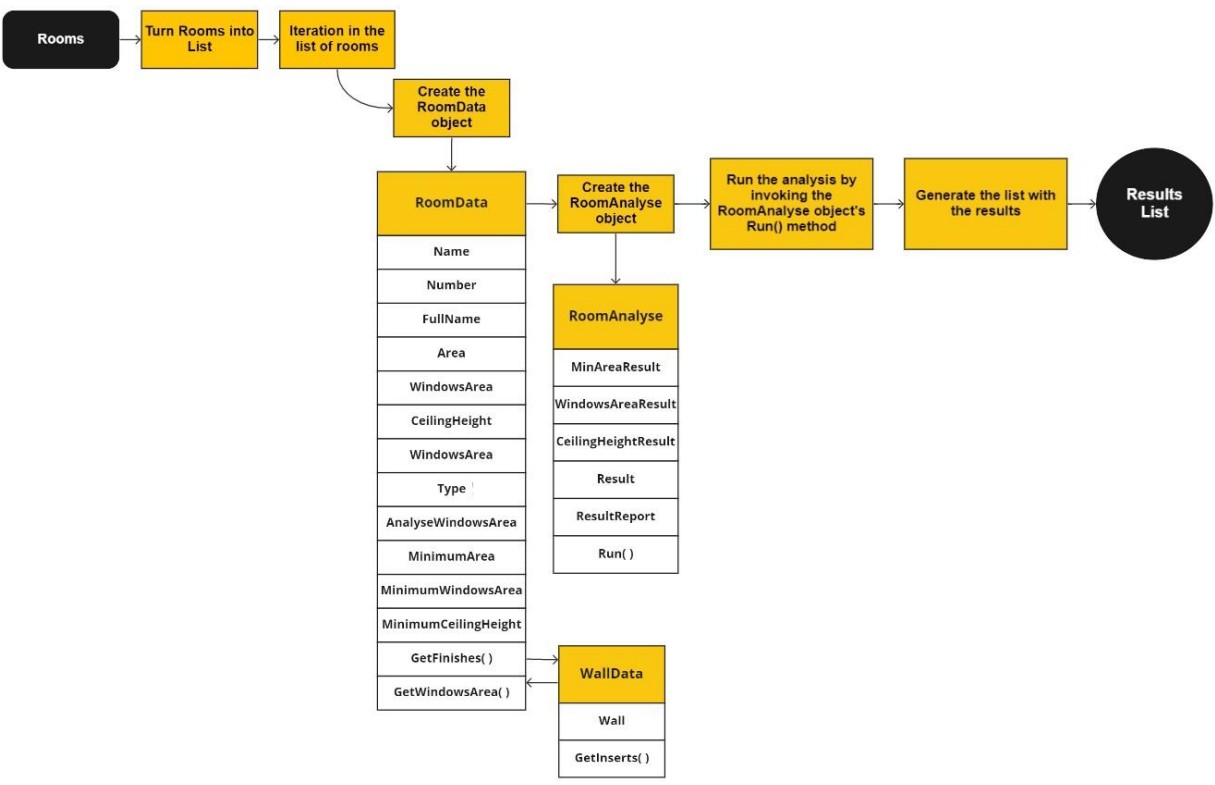

**Figure 5.** Room data information flow.

To allow the building code compliance analysis, the template quickly generates a schedule of windows and rooms (Figures 6 and 7), which will be used in the Dynamo routine to compare with the building code minimum requirements. Within these two schedules, all the information for the building code analysis is listed, including the dimensions and quantity of windows, the room's area and ceiling height.

| Type Mark | Width | Height | Sill Height | Quantity |
|---|---|---|---|---|
| J1 | 0.80 | 0.60 | 1.50 | 1 |
| J2 | 1.50 | 1.20 | 1.10 | 4 |
| J3 | 1.20 | 1.20 | 1.10 | 1 |
| **Name** | **Room Type** | **Unbounded Height** | | **Area** |
| Kitchen | Kitchen/dining room | 2.50 | | 8.26 m² |
| Bedroom | Bedroom | 2.60 | | 8.91 m² |
| Bathroom | Social Bathroom | 2.40 | | 3.19 m² |
| Bedroom | Bedroom | 2.60 | | 10.50 m² |
| Living room | Living room | 2.70 | | 12.98 m² |
| Laundry | Laundry | 2.70 | | 2.99 m² |
| Balcony | Undefined | 2.70 | | 4.20 m² |
| Hall | Hall | 2.70 | | 1.27 m² |
| Dining room | Kitchen/dining room | 2.70 | | 11.76 m² |
| Garage | Garage | 3.00 | | 32.40 m² |

**Figure 6.** SF building window and room schedule.

| Type Mark | Width | Height | Sill Height | Quantity |
|---|---|---|---|---|
| J1 | 200.00 | 100.00 | 110.00 | 46 |
| J2 | 100.00 | 80.00 | 160.00 | 20 |
| **Name** | **Room Type** | **Unbounded Height** | | **Area** |
| Balcony | Undefined | 270.00 | | 13.88 m² |
| Bathroom | Social Bathroom | 270.00 | | 4.03 m² |
| Bedroom 01 | Bedroom | 270.00 | | 13.14 m² |
| Bedroom 02 | Bedroom | 270.00 | | 8.37 m² |
| Hall | Hall | 270.00 | | 2.70 m² |
| Kitchen | Kitchen/dining room | 270.00 | | 12.35 m² |
| Laundry | Laundry | 270.00 | | 4.77 m² |
| Living room | Living room | 270.00 | | 31.14 m² |

**Figure 7.** MF building window and room schedule.

The template was previously characterised by the local building code minimum requirements for each type of compartment (based on Table 2), which were organised into a schedule (Figure 8). The Dynamo routine will later use the requirements from Figure 8 to compare with the BIM model characteristics (based on Figures 6 and 7). The results certify if the BIM model is complying with local building code regulations. If the BIM model is not approved, it indicates which rule is not being fulfiled. Whenever a room is not required to fill any of these requirements, the "Key Name" must be settled as "Undefined". Additionally, if a compartment is not required to have windows, the user can unselect it and exclude it from the analysis. Note that the designer must prepare the project and name each project compartment according to the names defined in the local building code (and in the Autodesk Revit template).

| Key Name | AnalyseWindowsArea | MinimumArea | MinimumWindowsArea | MinimumCeilingHeight |
|---|---|---|---|---|
| Bedroom | ☑ | 7.00 m² | 0.17 m² | 2.60 |
| Garage | ☑ | 10.35 m² | 0.05 m² | 2.30 |
| Hall | ☑ | 1.00 m² | 0.00 m² | 2.30 |
| Kitchen/dining room | ☑ | 4.50 m² | 0.13 m² | 2.30 |
| Laundry | ☑ | 1.60 m² | 0.10 m² | 2.30 |
| Living room | ☑ | 10.00 m² | 0.17 m² | 2.60 |
| Pantry | ☑ | 1.60 m² | 0.13 m² | 2.60 |
| Service Bathroom | ☑ | 1.60 m² | 0.13 m² | 2.30 |
| Social Bathroom | ☑ | 2.50 m² | 0.13 m² | 2.30 |
| Undefined | ☐ | 0.00 m² | 0.00 m² | 0.00 |

**Figure 8.** Building code requirements per compartment for Vila Velha and Florianópolis.

After the model characterisation and schedules organisation, the Dynamo routine was accordingly developed and it is presented in Appendix A, Figure A1. Overall, it captures the information from the BIM model and performs an automated action to validate the model according to the master plan indexes requirements. In addition, it uses the information from the template schedules to verify the model's compliance with the municipalities building code. The routine brings results in a few seconds, and it is divided into lines, each one corresponding to one requirement:

Number of levels: the first line of the routine concerns the analysis of the maximum number of levels for the building. As stated earlier, for a residential building located in the Priority Occupation Zone 03, the maximum number of levels is 15 and for AMC 12.5 Zone, the maximum number of levels is 10. This value is introduced as input for Dynamo to start the routine and then faced with the building number of levels (1 level for the SF building and 7 levels for the MF building), which was automatically collected from the model by Dynamo, bringing a positive result for this index.

Maximum height: the second flow line regards the analysis of the maximum building height. As in the previous requirement, the maximum height (47.00 m for Vila Velha and 45.00 m for Florianópolis) serves as an input for the Dynamo routine. Then, Dynamo will automatically identify the highest point of the building roof elements, and measure the distance to the site level, performing the comparison between the master plan requirements and the building height. Once again, both case studies have been approved for this index (SF building 4.05 m; MF building 20.30 m).

Utilisation coefficient, occupancy rate and permeable rate: the following stage from the Dynamo routine (third and fourth lines), assesses all the site-related indexes, namely the utilisation coefficient, the occupancy rate and the permeable area. As referred to earlier, users must define the site boundaries, the construction area and the permeable area. This is a mandatory step, as the Dynamo routine will use such information to perform the required calculations to assess all the site-related indexes. Note that a conversion node was introduced in the Dynamo routine to isolate the area values, as the input for Dynamo had both the area name, value and units (e.g., "Building area: 141.21 m$^2$" was converted to "141.21"). The calculation formulas were introduced in Dynamo and automatically performed to check the building's compliance with the master plan requirements. Results have shown positive feedback for both case studies. The SF building construction site presented a utilisation coefficient of 0.26 (less than the mandatory 3.50), an occupancy rate of 36.07% (less than 60.00%) and a permeable rate of 50.45% (over the minimum limit of 15.00%). The MF building construction site has also been approved with a utilisation coefficient of 2.16 (less than 4.80), an occupancy rate of 37.93% (less than 50.00%) and a permeable rate of 41.87% (less than 70.00%).

Front, back and lateral offset: the fifth line of the Dynamo routine concerns the building offset from the site boundaries. With a similar process to the previous indexes, the minimum requirements were given to start the Dynamo routine (Vila Velha's master plan—3.00 m for the front offset, 3.35 m for the back offset and 2.85 m for the lateral offset; Florianopólis master plan—4.00 m for the front offset, 1.50 m for the back offset and lateral offset). Then, the Dynamo routine was adapted to collect the distance between the building façades with the site boundary (defined by the user) and face them with the regulation limits. As presented in Figures 3 and 4, both case studies have been approved for these indexes, as they present the minimum required distances to their boundaries.

Slope of ramps: the last index from the municipalities master plans regards the slope of existing ramps, both for pedestrians and vehicles. This analysis is made in the routine sixth line, to check if the model ramps comply with the accessibility requirements. Under the Ramps category within Autodesk Revit, the user must fill the "slope" shared parameter in a percentage value, which later will be faced with the established maximum limits through Dynamo. This shared parameter was created in the Autodesk Revit template to facilitate data collection from Dynamo. Once again, both models have been approved for this index. The SF building exterior vehicle ramp (1.39%) and pedestrians' ramp (8.33%) have acceptable slopes, lower than the national limits of 20.00% and 8.33%, respectively. The MF building has also been successfully approved, as the existing vehicle ramp has a 15.00% slope (less than the 20.00% limit).

Minimum window area, room area and ceiling height: the last line from the Dynamo routine concerns the building code analysis, namely the minimum window area, room area and ceiling height. This analysis is made by comparing the schedules created by the used template (Figures 6 and 7), with the local minimum requirements of the building code, presented in Figure 8. To perform the analysis, the Dynamo routine checks if the room must be analysed for all the criteria, according to the user selection. Then, Dynamo gets the information of each building room to identify the dimensions and quantity of windows and to find the total window area of each room. The same process is made for the remaining criteria, by identifying the area of each room, as well as the respective ceiling height. Finally, the building results are faced with the building code requirements to check if the building can be approved or not for construction. For the specific SF building case study, all the criteria were accomplished with an exception for the minimum window area, which was not reached in some of the building rooms—the laundry does not have windows, while the minimum requirement is to have at least 0.10 m$^2$ of windows. The MF building case study has been approved for all the indexes. Note that the Dynamo routine associates the room names with the "key names" of Figure 8, so the user must name rooms accordingly.

The routine output consists of a compliance result, stating if the building can be approved by the city hall, showing which parameters have not been approved according to the considered regulations. Figure 9 presents the final Dynamo node—compliance report—where the SF building case study has failed in complying with all the applicable regulations, namely the building code requirements (red circles). As one of the research goals was also to provide a decision support tool for designers, it was necessary to present the result for each index and criteria in an isolated manner. The "code block" node presents six outputs (green circle), each one corresponding to the analysed indexes, as presented earlier in this section—number of levels; maximum height; utilisation coefficient, occupancy rate and permeable rate; front, back and lateral offset; slope of ramps; minimum window area, room area and ceiling height. A "watch" node can be introduced in each output to quickly identify the analysis results for each index. This way, designers can quickly understand where the building design is failing and take action to improve it.

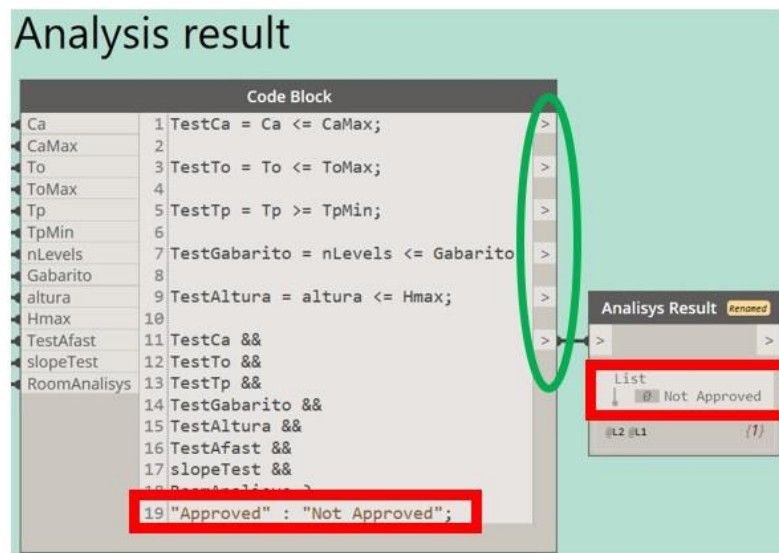

**Figure 9.** SF building analysis result.

However, some of the compliance report outputs aggregate the results from several indexes, namely:

- Output 3 aggregates the utilisation coefficient, the occupancy rate and the permeable rate.
- Output 4 aggregates the front, the back and the lateral offsets.
- Output 6 aggregates the minimum window area, the minimum room area and the minimum ceiling height.

Whenever these outputs provide a negative result and the user needs to identify which criteria are not being approved, he must check the routine lines individually. For example, the SF building case study has failed in output 6, which concerns the building code criteria. To identify which specific criterion was not complying, the user must check the respective routine line, as presented in Figure 10. Thus, it was possible to identify that the minimum window area criterion was not being complied with for one of the building rooms—the laundry (red circle).

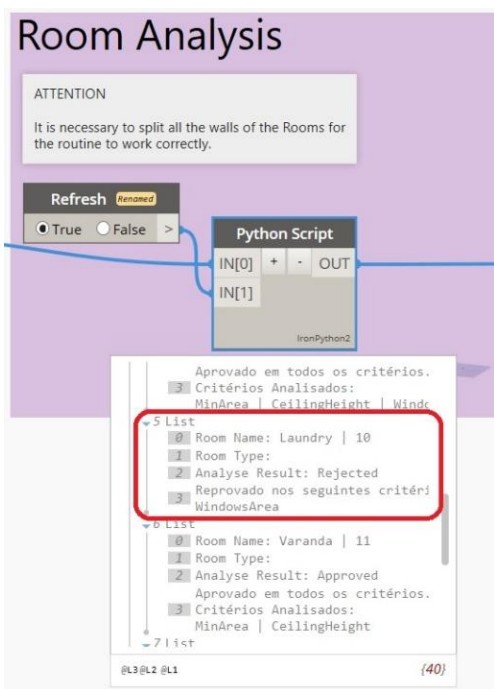

**Figure 10.** Individual output analysis (SF building).

## 4. Discussion

Using the BIM models created in Autodesk Revit, together with a personal template and the routine developed in Dynamo, it was possible to analyse and automate the verification of urban indexes compliance for different case studies in Brazil. The execution of the automated routine was significantly faster than the typical analysis process and has provided reliable results. Using the traditional index evaluation method, human errors were usual both in assessing building data and when performing the index calculations. Such issues were common among designers, which were required to guarantee the building compliance before project submission, but also for municipality evaluators, which must verify all the rules and indexes compliance (double work). Besides the considerable time required, such errors could also contribute to delaying all the process, creating a less efficient working method. The main advantages of the automated routine concern the precision, quickness and reliability of the results, leading to faster and more efficient analysis, both for designers and municipalities. Moreover, such a routine can also significantly support designers' decisions when creating innovative design options, as they can have real-time information about the project design compliance.

The case study results have proved the method's applicability in different buildings and regions of Brazil. Both models have been successfully assessed. The SF building has not been approved to issue a building permit, as it failed in complying with the minimum window area requirements. Understanding this constraint during the project's early stages can support designers in selecting new solutions to comply with local and national regulations, with low efforts and costs. Verifying such issues near to project submission (traditional practice) would eventually lead to major changes in project design—updating architectural and engineering project documentation, client approval and material selection, among others—which entails higher costs and considerable time. The MF building has been quickly assessed and approved, only by updating the inputs for the Dynamo routine—Florianopólis index limits—as presented in Figure 11. The automated analysis has provided real-time compliance results, allowing to optimise resources and efforts and improve project efficiency. City hall experts would also have similar benefits, as well as reduced bureaucracy and faster project appraisal.

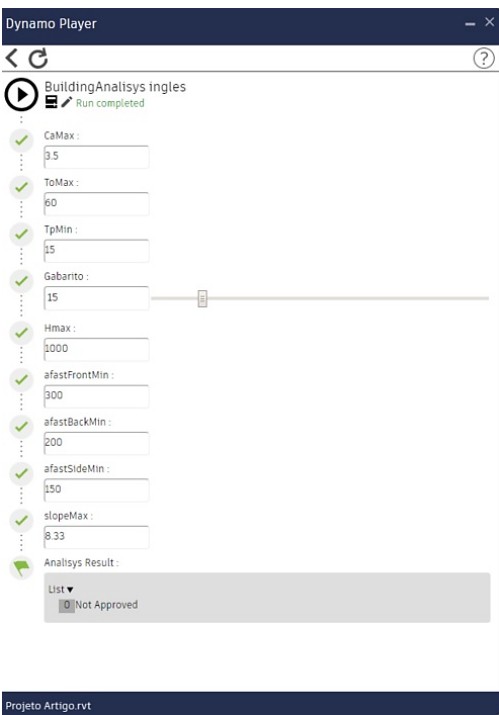

**Figure 11.** Input/output in Dynamo Player.

The case studies have also demonstrated that the workflow created with Autodesk Revit and Dynamo must follow specific modelling rules, for a proper data collection from the BIM model. If the modelling stage does not follow such guidelines, specific errors may occur during the analysis. As an example, if the building roof is modelled under the "Floor" category, the Dynamo routine will not be able to assess the maximum building height index, as it considers the highest point of slabs modelled under the "Roof" category. Similar problems may happen if walls are not divided in their intersections, making unfeasible the identification of which walls are associated with each building room. In addition to the modelling guidelines, a personal template is also required, which has been characterised by a set of shared parameters and predefined schedules for facilitating and fostering data collection. The use of such a template is mandatory for the analysis, as the Dynamo routine specific uses the schedule data to perform the calculations. The template can be easily used and replicated, only by updating the building code indexes (Dynamo inputs), according to the project type and location. The template use can be avoided by creating a second Dynamo routine, which would perform the same actions, but would also create the need to run such a routine before the modelling stage. Moreover, the use of the applied method will always require the creation of a BIM model, which is mandatory for the designer assessment, as well as for the city hall evaluator.

Another key factor for the Dynamo routine functionality is the use of Autodesk Revit 2022 (or a more recent version) in English, and a Dynamo 2.10 version (or higher). This limitation appears because each year the software has new updates that can compromise the created routine, so it must be carefully reviewed at each update, preventing it from not working. However, it is expected that the routine will work properly in more recent versions, as only basic nodes were used.

When concerning the Dynamo routine limitations, two major factors stand out:

- The need to check the routine lines for assessing which index is not being complied—as stated earlier, some indexes are assessed together (output 3, 4 and 6 from Figure 9). If some of these indexes are not being approved, the user must carefully check the routine lines to assess which one is not complying with the local standards. To overcome such an issue, the Dynamo routine should be extended and isolate all the index assessments.

Note that this procedure will slow down the assessment, as the routine should be completed and will have a set of new coding lines to process.

- The compliance report presentation—up to date, the analysis detailed result must be assessed directly in Dynamo, requiring the need to open the routine to assess results. There is no need to perform any action in the routine but for unfamiliar users, it may take a while to understand Dynamo logic, hindering its potential use. Nevertheless, the global compliance result (output of Figure 11) may be seen directly in Dynamo Player without any effort. To match this need, a node from "datashapes" [23] can be used, which allows presenting a given result or information directly in the Autodesk Revit environment, only by running the routine with Dynamo Player. Such nodes can present both the individual result of each index and the building's final evaluation.

Following these considerations and the case study results, the method's functionality and replicability have been proven for Brazilian locations and buildings. For further use in other Brazilian cities, the user must only introduce the local master plan indexes to start the Dynamo routine (Figure 11 inputs), according to the building type and location. For residential building purposes, the Autodesk Revit template containing the local building code requirements can be directly used, as limits will be kept constant (only for Brazilian residential buildings). For other building types, the template limits must be updated directly in Autodesk Revit with the new requirements (if needed). The replication for other countries can face an additional constraint, as indexes and requirements may be different. Nevertheless, the methodology concept and Dynamo routine can be easily adapted, only by updating the indexes or adding new ones directly into the routine. Data collection, processing and presentation will remain identical, as well as the whole methodology concept.

## 5. Conclusions

In this study, the workflow and the creation of automated routines using Autodesk Revit and Dynamo software were evaluated, exploring its potential to support designers in assessing their project's compliance with local and national regulations. With this method, a novel procedure was established for automatic verification of the Brazilian master plan and building code indexes. It provides real-time outputs during the project's early stages, allowing design innovation and project changes with few resources. To demonstrate the concept, two case studies were evaluated and it was proven that the use of BIM provides the required resources for a faster and more reliable data collection and analysis. Speed, consistency and efficiency in assessing building compliance were demonstrated. It has provided important data for decision-making during the design phase, but also for municipalities to quickly assess the building's compliance with the local regulations and reduce process bureaucracy.

It was observed that Autodesk Revit has the potential to store the required multi-disciplinary information for the project evaluation, while Dynamo has the capacity to gather such data and perform the required calculations. Moreover, every BIM model characteristic can be assessed and processed by Dynamo, allowing for the creation of different types of quantitative and/or qualitative assessment routines. Despite the existence of some limitations, the method has proved to be quite reliable, offering a couple of options to overcome them. Overall, it has been concluded that the use of this software has the potential to become an essential tool both for the dissemination of BIM, as well as for the compliance assessment automation. It can significantly support design, as well as reduce the administrative processes and time of getting construction permits.

Finally, it is noteworthy that the applied methodology can be directly replicated for other Brazilian buildings and locations, as well as for other countries by performing punctual adaptations to meet the standards of each location.

**Author Contributions:** Conceptualisation, F.S.V. and J.P.C.; methodology, F.S.V.; software, F.S.V.; validation, J.P.C., F.S.V. and L.B.; formal analysis, F.S.V. and J.P.C.; investigation, J.P.C. and F.S.V.; writing—original draft preparation, F.S.V. and J.P.C.; writing—review and editing, F.S.V., J.P.C. and L.B.; visualisation, J.P.C.; supervision, L.B.; funding acquisition, J.P.C. All authors have read and agreed to the published version of the manuscript.

**Funding:** This research was funded by the Portuguese Foundation for Science and Technology, through the Regional Operation Programme of North (Grant number SFRH/BD/145735/2019).

**Institutional Review Board Statement:** Not applicable.

**Informed Consent Statement:** Not applicable.

**Data Availability Statement:** Data is contained within the article.

**Conflicts of Interest:** The authors declare no conflict of interest.

**Abbreviations**

| | |
|---|---|
| AMC | "*Área Mista Central*", Central Mixed Area |
| BIM | Building Information Modelling |
| LOD | Level of Development |
| MF building | Multi-Family Residential Building |
| SF building | Single-Family Residential Building |
| VPL | Visual Programming Language |

**Appendix A**

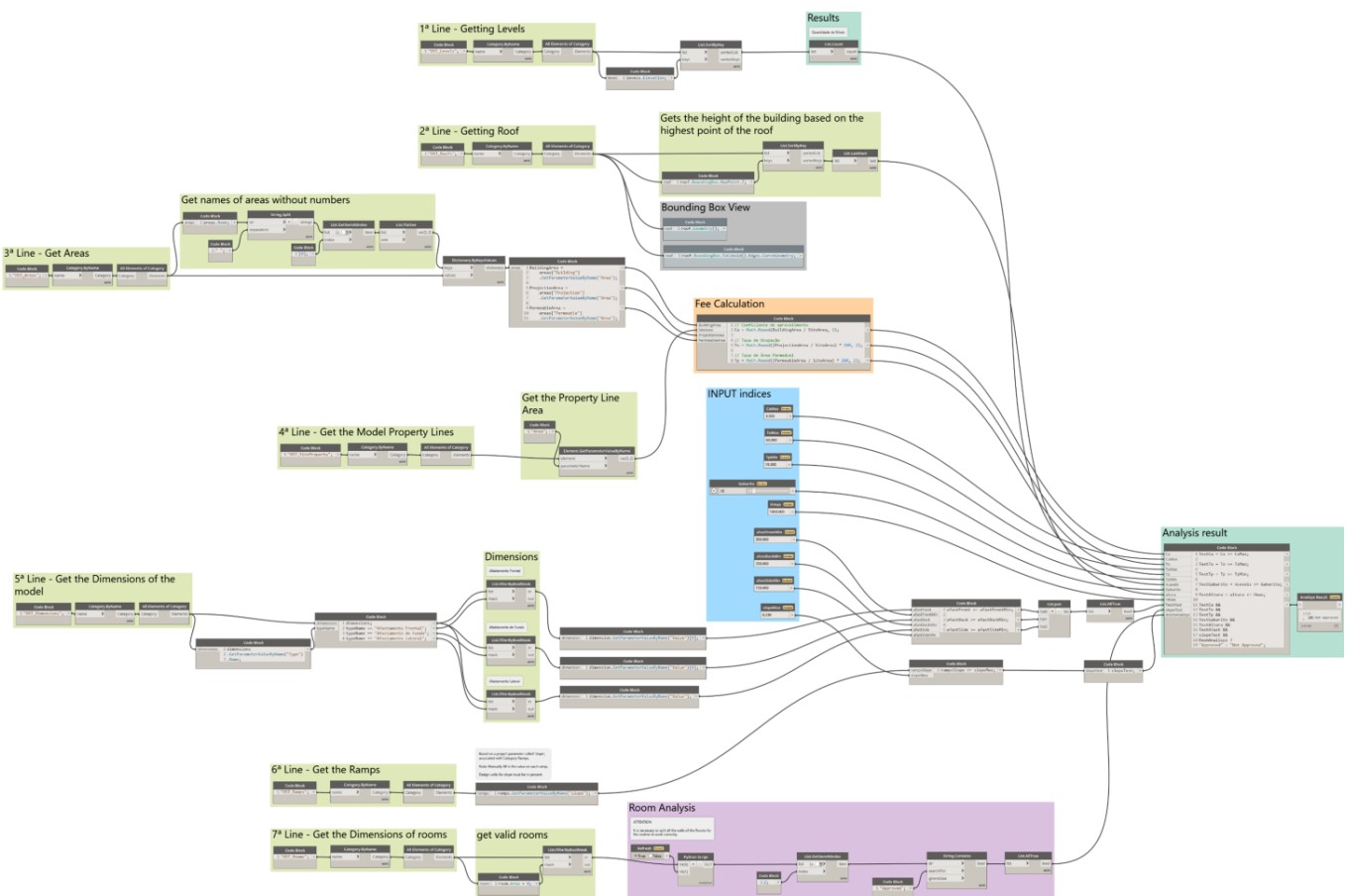

**Figure A1.** Dynamo routine.

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
