# Peer review of "BIM-Based Method for the Verification of Building Code Compliance"

_asi, doi:10.3390/asi5040064_

Round 1
Reviewer 1 Report
(1) The paper describes the development of a BIM-based method (or algorithm) for the verification of the code-compliance of buildings. In particular, Autodesk Revit was used to develop the BIM model of the building, whereas the assessment of the code-compliance of the BIM model was carried out by means of software Dynamo. The code rules of Vila Velha (Brazil) municipality were considered as a reference, and the case study building consisted in a one-level residential building, which is relatively typical of the reference urban area.
(2) The paper is well-structured and well-written, and English style is appropriate. The paper potentially contributes to practice and professional applications in the field of civil engineering and construction technology, rather than to the scientific research. In particular, the developed method or algorithm might be considered as a reference, or might be further developed, towards the enhancement of the efficiency of both design and compliance check processes, also minimizing the influence of human error on these processes. However, the referee is not completely certain that the paper can be considered as an original scientific contribution, for the relatively reduced contribution in terms of originality and innovation within the relevant research and science scopes. As an additional comment, it was noted that a single case study was considered in the study for the application of the method, and this might represent a potential issue in terms of representativity and robustness of the study.
(3) In the light of the abovementioned review comments, the Authors are kindly asked to provide (as a review response and within manscript) robust motivations and possibly, evidence, in order to prove that the manuscript merits publication in ASI. In particular, it should be reasonably demonstrated that the contribution of the paper fills a specific (clearly identified) research gap and that the paper is significant in terms of research and science. Furthermore, the Authors are strongly suggested to consider additional case study implementations and further application examples.
Reviewer 2 Report
The article concerns a unit solution in Vila Velha, Brazil. For this reason, the title seems too general and too promising. Especially that the authors stated (lines 482-483) that "The replication (of the method) for other countries 482 can face some more constraints, as indexes and requirements are usually different". The use of two existing tools (Autocad Revit + Dynamo) is a good suggestion, although the reviewer sees insufficient reflection on the automation of the design process. It should be noted that the process of automating design is highly up-to-date and has a wealth of literature (see, e.g., N. Leach, Architecture in the Age of Artificial Intelligence: An Introduction to AI for Architects or proceedings from the Conference eCAADe2018). The proposed method fits into a broader context of using tools optimizing the work of an architect-town-planner; therefore, enhancing the discussion section would be valuable.
The proposed method might be suitable for the analyzed example. Still, its implementation for other areas that require a qualitative assessment (such as cultural heritage or natural environment protection zones) is doubtful because some of the requirements of municipality urban plans cannot be recorded using indicators.
The proposed method may be useful, but the manuscript needs to be supplemented as indicated above in the opinion of the reviewer.
Round 2
Reviewer 1 Report
The referee is satisfied with the authors' answers and with the reviewed version of the manuscript, which improved in quality. The referee's concerns associated with the potential lack in scientific contribution have been dissipated. Therefore, the paper is now suitable for publication.
